# Sonmat, a citizen-science enabled Kimjang kimchi case study on associations between hand and kimchi microbiota

Wannes Van Beeck,[1] Tom Eilers,[1] Wenke Smets,[1] Lize Delanghe,[1] Dieter Vandenheuvel,[1] Ines Tuyaerts,[1] Joke Van Malderen,[1] Sarah Ahannach,[1,2] Katrien Michiels,[1] Caroline Dricot,[1] Nele Van de Vliet,[1] Ae Jin Huys,[3,4] Patrick De Boever,[5,6,7] Sarah Lebeer[1,2]

**ABSTRACT**  Kimjang kimchi is traditionally made in Korea in autumn to preserve vegetables during colder winter times after the harvest. Kimjang is an important societal tradition in which families and communities come together to process vegetables, such as cabbage, into kimchi. The origin of the microorganisms that contribute to the flavor and safety during fermentation is still unclear. Although bacteria present on the raw ingredients are considered to be important colonizers of the fermentation, in Korean culture, the term "Sonmat" is often used, which literally translates into "hand flavor," suggesting a role for hand microbiota in the kimchi fermentation. In this citizen-science project, we investigate the impact of the hand microbiome on kimchi fermentation during the Sonmat festival organized in Belgium. The kimchi fermentations contained mainly lactic acid bacteria belonging to the genera *Leuconostoc*, *Weissella*, and *Latilactobacillus*. The hand microbiota was characterized by the presence of *Staphylococcus*, *Corynebacterium*, *Micrococcus*, and *Enhydrobacter*. Associations were found between the relative abundance of *Staphylococcus* on the hand and the relative abundance of *Latilactobacillus* and *Leuconostoc* found in kimchi, despite limited overlap between the hand and the kimchi microbiome. In addition, different microbiota were found to dominate the kimchi made following the traditional group Kimjang practices compared with individually prepared kimchi. These findings pave the way for future research into how traditional practices and the skin microbiome influence the unique qualities of kimchi, offering exciting possibilities for enhancing fermentation processes and cultural food heritage through citizen science.

**IMPORTANCE**  Vegetable fermentation has been a staple of human culture for centuries, with deeply rooted traditions behind it. However, the effects of these traditional practices on the microbes in the final fermented product, and their origin, are often not understood. By using participatory citizen-science approaches, it is possible to study these important foods while preserving the authenticity and integrity of the traditional fermentation practices that define them. The results obtained from our citizen-science case study support the importance of exploring traditional fermentation practices and their effect on microbial and sensory properties of fermented foods. Additionally, our case study found associations between microbiota present on the hand and microbiota important in the early successional stage of kimchi fermentation.

**KEYWORDS**  Kimchi, fermentation microbiota, citizen science, skin microbiota, Kimjang, fermented food

Address correspondence to Sarah Lebeer, sarah.lebeer@uantwerpen.be.

S.L. received funding from different probiotic companies that were not involved in this research. S.L. is also an academic board member of ISAPP, which was not involved in this work. L.D. was affiliated with the university at the time of the study. Her salary was funded by VLAIO through a Baekeland mandate in collaboration with YUN NV, which is not involved in this research. T.E. is partially funded through an industrial research VLAIO grant not related to this work. A.J.H is curator of sonmat and curator and owner of Mokja (www.mokja.be). The remaining authors have no conflicts of interest to declare.

See the funding table on p. 12.

Kimchi is a traditional product produced in Korea consisting of fermented vegetables such as cabbage, radish, cucumber, and spices, including red pepper powder, salt, garlic, ginger, fish sauce, and many other ingredients (1). The word "kimchi" originated from "chimchae," the Chinese character, meaning salted vegetables. It was first called

"dimchae" or "dimchi," which evolved into kimchi through the word of mouth (1). The origin and exact date of discovery of kimchi is highly debated. However, the earliest reference to fermented vegetables can be found in the Sikyung (book of odes), which was published in 500 BC. Here, the term jeo (菹) was introduced, which is believed to refer to fermented vegetables and eventually kimchi (2).

Traditionally, salt and a (semi-)anaerobic environment in closed jars are used to encourage spontaneous fermentation, which results in the creation of kimchi (3). Several studies have examined the microbiota present in kimchi through culture-dependent (4–6) and culture-independent approaches (7–10). In general, a microbial succession occurs, similar to other brine-based vegetable fermentations, with first a community native to the fresh vegetables, which typically contains members of the *Enterobacteriaceae,* then shifting toward a community rich in lactic acid bacteria (LAB) (11). LAB genera, such as *Weissella, Leuconostoc, Pediococcus, Lactiplantibacillus,* and *Latilactobacillus,* are often found in these fermentation communities (12, 13). LAB have a competitive advantage compared with unwanted bacteria such as many *Enterobacteriaceae* due to their high salt tolerance and their capacity to convert sugars into organic acids (14). Fresh produce is better preserved because of the generation of acids during the fermentation process, which lowers the pH of the surrounding environment in the fermentation jar and prevents the growth of spoilage and possible foodborne Enterobacterales organisms like *Salmonella enterica* Typhimurium or *E. coli* O157:H7 (15). Overall, there are more than 100 different varieties of kimchi described, based on regional ingredients and traditional practices, which are often handed down from generation to generation (16).

Kimjang (pronounced Gimjang) is one of these traditional practices, occurring in Korea during autumn, which dates back thousands of years ago (17). The last vegetables harvested in autumn before the winter are processed together by multiple households in the community. The salted shredded vegetables are traditionally stored in large earthenware jars called *hangari* or *onggi*, which have a porous structure that makes them permeable for gases, but not water (18). These jars are traditionally buried underground to achieve a slow fermentation process to enable the Kimjang kimchi to be available throughout the whole winter (18). Nowadays, often plastic containers are used to store and ferment kimchi. In 2013, the Kimjang tradition was embedded in the Representative List of Intangible Cultural Heritage of Humanity by the UNESCO (2, 18).

The general microbial community dynamics of vegetable fermentations such as modern cabbage kimchi are rather well studied (11, 19–21), but the precise origin of the microbiota that colonize the fermentation is not yet fully understood. Culture-independent approaches have not been used to investigate the traditional practices on the kimchi's underlying microbial community. Besides bacteria native to the fresh produces, bacteria present on the hands of the person(s) making kimchi are hypothesized to be a potential source of microbes. In sourdough, the effect of the hand microbiome on the sourdough starter has already been shown (22). In Korean culture, the term "sonmat," sometimes also extended to umma sonmat, refers to the mother's hand flavor and to the inherited quality, love, and care that went into preparing the dish (23). As in sourdough, it can be speculated that the hand skin microbiota plays a role in sonmat, but this has—to the best of our knowledge—not yet been investigated. The human skin microbiota is an important first barrier against pathogen invasions and considered one of the largest organs of our human bodies (24). The skin harbors a diverse microbial community with several commensal members of genera such as *Staphylococcus*, *Corynebacterium, Cutibacterium,* and the family of the *Lactobacillaceae* (25). Thus, the microbiota of the skin could theoretically be a potential source of LAB that can colonize and persist within fermented food, but this remains to be substantiated.

Citizen science engages laymen for sampling and scientific analysis activities (26, 27). Of note, community science is also gaining traction as a more inclusive alternative; however, we believe that it is best reserved for initiatives that are both driven by and directly benefit specific communities, which may not apply to all citizen science projects. Citizen science ranges across different participatory levels ranging from minimal

involvement to co-creation (27). It has previously been implemented for microbial analyses of fermented foods, mainly using the second participatory level of sampling, for fermented carrots (28), kefir (29), and sourdough (30). Citizen science has also been used for microbial analyses of the human microbiome, such as in American gut microbiome project (31, 32) and Isala project on the vaginal microbiome (33), but studies linking human skin microbiota and fermented foods in a citizen science context are scarce. In this study, we combined the analysis of fermented food (kimchi) and the human skin microbiota within a citizen-science festival workshop during which citizens, that is, the organizers of the Sonmat festival including chefs and stakeholders, also had the opportunity to contribute to ideas for the fermentation workshop and set-up during a design brainstorm. We investigated the effect of skin microbiota and traditional Gimjang practices on the microbial community in a cabbage-based kimchi fermentation while controlling for hand disinfection prior to preparation, raw material source, processing conditions (e.g., chopping), and storage conditions. Two workshops were organized: one workshop during which participants individually made kimchi and one traditional Kimjang group kimchi workshop during which participants made kimchi together in a group (Fig. 1). The microbial communities of the hands of the participants (in the individually made kimchi workshop) and the resulting kimchi were analyzed and compared. The results were communicated back to the participants after the conclusion of the project. This case study aimed to generate new insights and hypotheses regarding the origin of bacteria in traditional vegetable ferments and to explore how traditional fermentation practices influence the microbial community within the fermentation process.

## RESULTS

### Kimjang group-made kimchi is more uniform in pH and gas formation compared with the individually made kimchi

First, we measured the pH as an important indicator for safety during the fermentation process. A pH of 4.6 is a safety threshold set by the food safety agency, below which no toxin production or growth of pathogens is expected (34). The pH of both individually made and Kimjang group-made kimchi was measured after 3 days of fermentation. Individually made kimchi samples reached a pH of 4.61 ± 0.28, and the group kimchi reached a pH of 4.29 ± 0.09 (Fig. 2). After 31 days, the pH of the group kimchi samples reached 3.80. Interestingly, a more heterogeneous pH was observed within the individual kimchi samples, indicating a higher interindividual variability than the group kimchi.

Gas production was also visually assessed as an indicator of microbial activity during the early stages of fermentation. Carbon dioxide ($CO_2$) production by heterofermentative LAB is commonly observed in plant-based fermentations (11). In line with the pH results, gas production was only observed in 36% of individually made kimchi samples, compared with 71% of the group kimchi samples.

### Kimjang practices result in a taxonomically different LAB dominance

The impact of the fermentation practices was examined through culture-independent RNA-based V4-16S rRNA gene sequencing to assess the viable community, with genus-level resolution. After 3 days of fermentation, the individually made kimchi had a higher alpha diversity (e.g., the diversity within samples) compared with the Kimjang group kimchi (Fig. 3C). Higher alpha diversity in the individually made kimchi could be attributed to an increased evenness in these samples, as no significant change in richness was observed between individually and group-made kimchi (Fig. 3C; Fig. S1). A significant difference in beta diversity using the Bray-Curtis index was observed between individually and group-made kimch (Fig. S2). In general, all the individual and group kimchi fermentations were dominated by LAB, reaching a relative abundance of over 50% in each of the fermentations. Within the individually made kimchi samples, two ASVs of *Leuconostoc* (1 and 2) were observed, which reached relative abundances of over 50% in

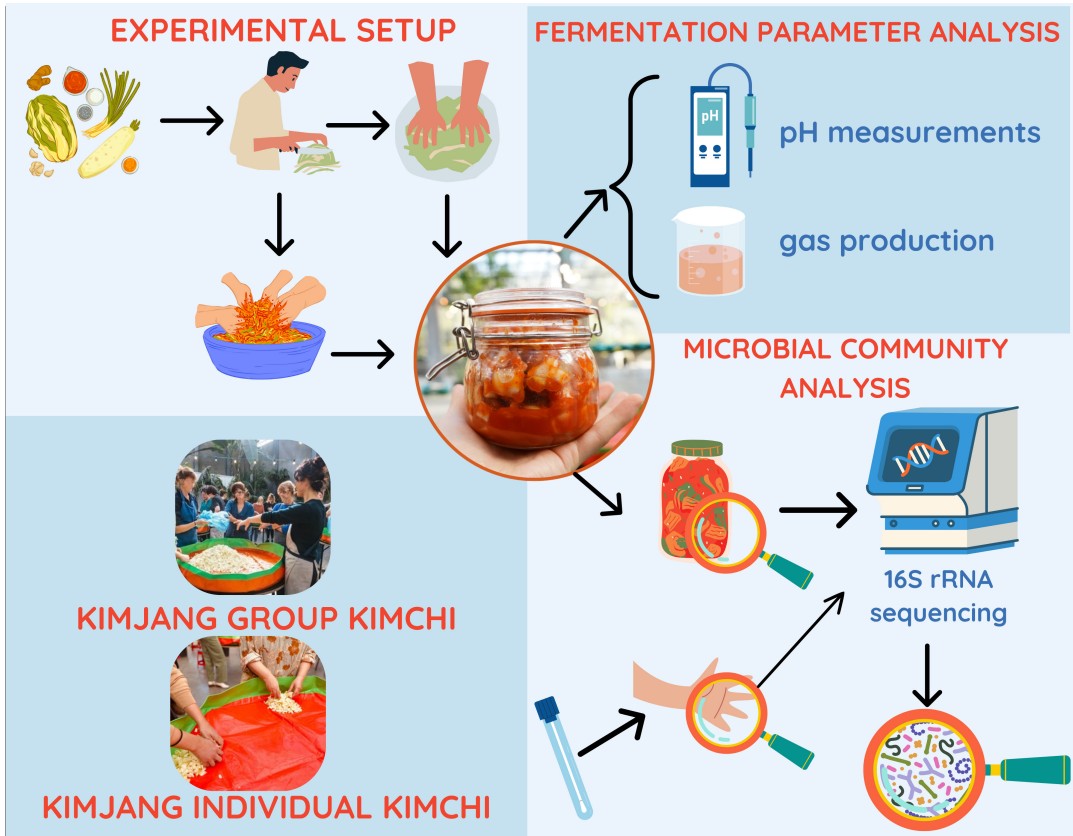

**FIG 1** Experimental overview of the Kimjang kimchi session during the Sonmat festival. Two sessions were organized. In the first session, participants made kimchi in a group (see the left-hand panel figure). During the second session, kimchi was prepared individually (the right-hand panel), and skin microbiota samples were taken from the hands before the start. Kimchi fermentations were followed up in the lab by observing gas production and measuring pH after 3 days. Microbial community composition was determined through V4-16S rRNA sequencing of both hand and kimchi samples, and overlap in bacterial taxa (at the level of amplicon sequence variants [ASVs]) was determined. Photographer—Willem Devriendt, Photogenica Fotografie

17 of the 25 fermentations studied. In the other fermentations that did not reach 50% of *Leuconostoc* relative abundance, a higher abundance of *Latilactobacillus* 1 was observed. A different community was observed for the Kimjang group-made kimchi fermentation. Seven of the eight fermentations studied were dominated (relative abundance >75%) by *Weissella*, whereas the other fermentation had a mixed community with *Weissella*, *Pediococcus,* and *Latilactobacillus* as key genera (Fig. 3A). *Leuconostoc* and *Latilactobacillus* were the most abundant genera present in the individually made kimchi, and *Weissella* was significantly more abundant in the group compared with the individually made kimchi (Fig. 3B for ANCOM-BC2 results and Fig. S3 for other differential abundance metrics).

The microbiota present in the fermentation were also correlated with fermentation parameters (gas and pH). We correlated the gas production and pH with microbial community composition at beta and alpha diversity levels. Both pH and gas production showed a significant correlation with the microbial community during kimchi fermentation, and 18.5% of the bacterial beta diversity within the fermentation could be explained by the pH variable (Bray-Curtis dissimilarity, $R^2$ = 18.5%, F = 6.85, *P* = 0.002), whereas 10.8% of the bacterial diversity within the fermentations could be explained by the presence or absence of gas production (Bray-Curtis dissimilarity, $R^2$ = 10.8%, F = 4.01, *P* = 0.025).

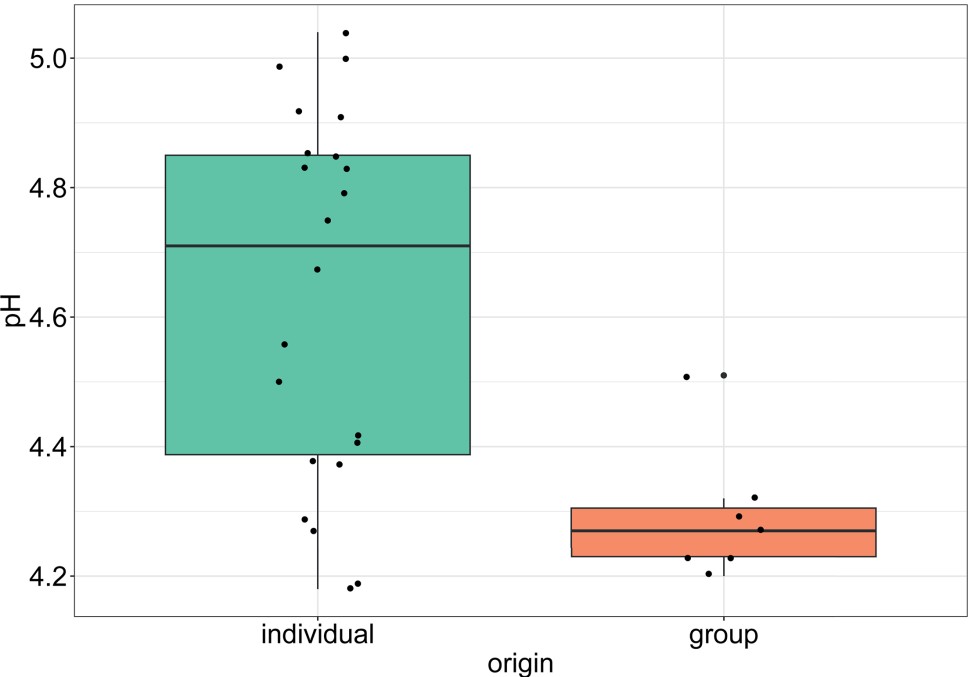

**FIG 2** pH of kimchi samples of individually made versus Kimjang group-made kimchi. The pH of the samples was measured after 3 days of fermentation. More variability and higher pH were observed for the individual kimchi samples compared with the Kimjang group kimchi.

## The hand skin has a distinct microbial community compared with the kimchi fermentation

The potential of the skin as a source of microbial colonizers of the fermentation ecosystem was then explored by analyzing participants' hands of individual kimchi. The hand microbiome was characterized by a higher alpha diversity compared to the fermentation microbiome (Fig. 4C; Fig. S4). A significant difference in beta diversity using Bray-Curtis index was observed between the skin and both fermentation microbiota (PERMANOVA, $P < 0.05$, Fig. S5). *Staphylococcus* was the most abundant genus found on the skin, followed by *Enhydrobacter, Corynebacterium, Acinetobacter,* and *Micrococcus* (Fig. 4A). Differential abundance analyses between skin and kimchi microbiome confirmed that *Staphylococcus, Pseudomonas,* and *Enhydrobacter* were significantly more abundant on the hand skin than in the kimchi fermentation, which was characterized by a significantly higher abundance of *Leuconostoc, Latilactobacillus,* and *Enterobacter* (Fig. 4B for ANCOM-BC2 results and Fig. S6 for other differential abundance metrics).

To explore the correlation between the hand microbiome and the kimchi microbiome in more detail, individual profiles of hand and fermented foods were matched per participant. Some overlap in microbial taxa shared between the hand microbiota and kimchi was found for 16 of the 19 participants, but this was overall limited, with a maximum of 6 ASVs shared between skin and kimchi of a single participant (Fig. 5A). Most ASVs that were shared between hand and kimchi fermentation belonged to *Leuconostoc, Staphylococcus, Enterobacter,* and *Enhydrobacter,* but only ASVs belonging to *Leuconostoc* and *Enterobacter* reached a relative abundance above 1% in the kimchi samples. *Staphylococcus* and *Enhydrobacter* had an average relative abundance below 1%; thus, they did not establish themselves within the fermentation communities. These observations suggest that microbiota originating from the skin can potentially persist in kimchi fermentation but do not dominate the microbial community. However, the presence of specific taxa on the human skin appeared to indirectly influence the microbial community in kimchi fermentations because the relative abundance of *Staphylococcus* on the skin was significantly associated with the microbial diversity

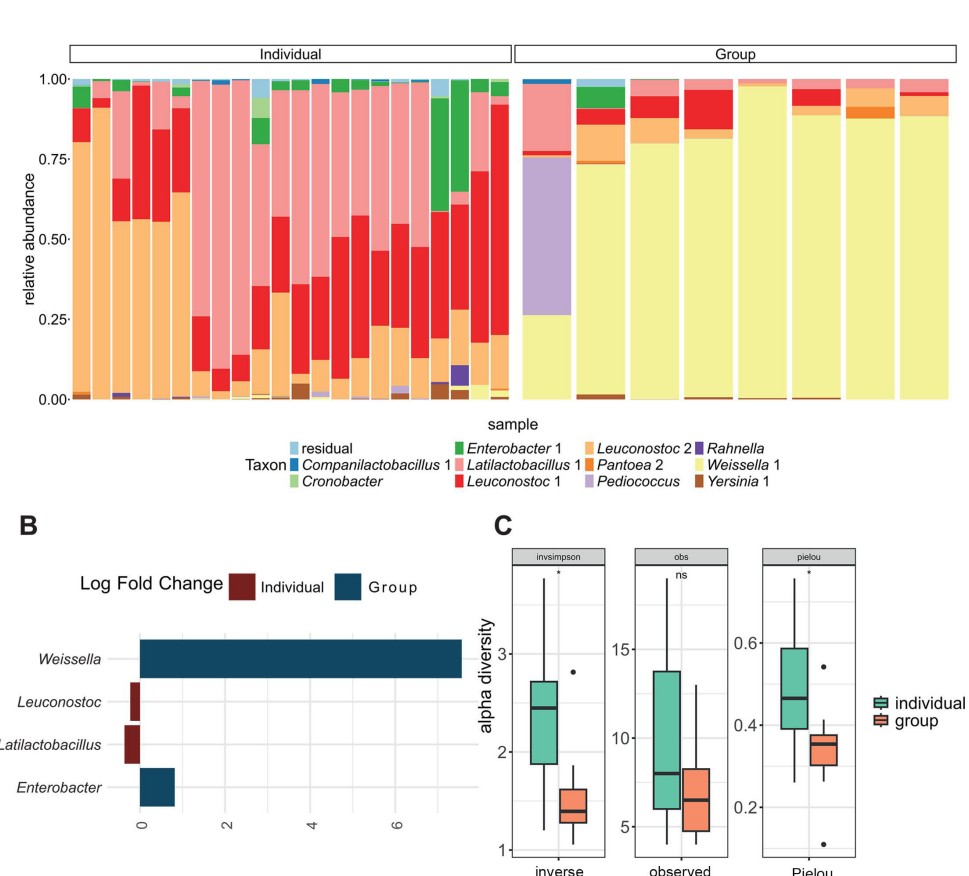

**FIG 3** (A) Bacterial community composition of individually made (left, $N = 22$) and Kimjang group kimchi (right, $N = 8$) after 3 days of fermentation. The bacterial community composition was determined using V4-16S rRNA sequencing. The top 11 most abundant ASVs were visualized in a stacked bar graph. (B) Differential abundance of microbiota between individually made and group-made kimchi microbiome. Differential abundance of genera with at least 1% mean relative abundance per method (individual or group) was calculated with ANCOM-BC2. Bars were colored based on the location where they had a significantly higher differential abundance. (C) Alpha diversity of individually made and Kimjang group kimchi samples after 3 days of fermentation. The alpha diversity was calculated using the inverse Simpson metric, which takes into account both richness and evenness of the microbial community, observed which takes into account only richness and Pielou, which is a metric for the evenness of the microbial community.

within individual kimchi fermentations ($R^2 = 15.0\%$, $F = 3.02$, $P = 0.047$). A negative correlation was also found between *Staphylococcus* on the hand and *Latilactobacillus* within the kimchi fermentation ($R = -0.42$, $P = 0.013$, Fig. 5B). Another negative correlation was found between *Staphylococcus* present on the hand and *Leuconostoc* within the kimchi fermentation ($R = -0.36$, $P = 0.034$, Fig. 5C).

## DISCUSSION

In this study, we explored the effect of traditional Kimjang (group-based) fermentation practices on the microbial dynamics during the fermentation. The Kimjang group kimchi microbiota was dominated by *Weissella* compared with more diversely dominated individually prepared kimchi. A more uniform distribution of pH for Kimjang group-made kimchi was also observed compared with individually made kimchi studied here. *Weissella* was highly abundant in the Kimjang group kimchi, whereas this genus was only a minor bacterial player in the individually prepared kimchi community. *Weissella* is a heterofermentative LAB that is commonly found in kimchi fermentations (35).

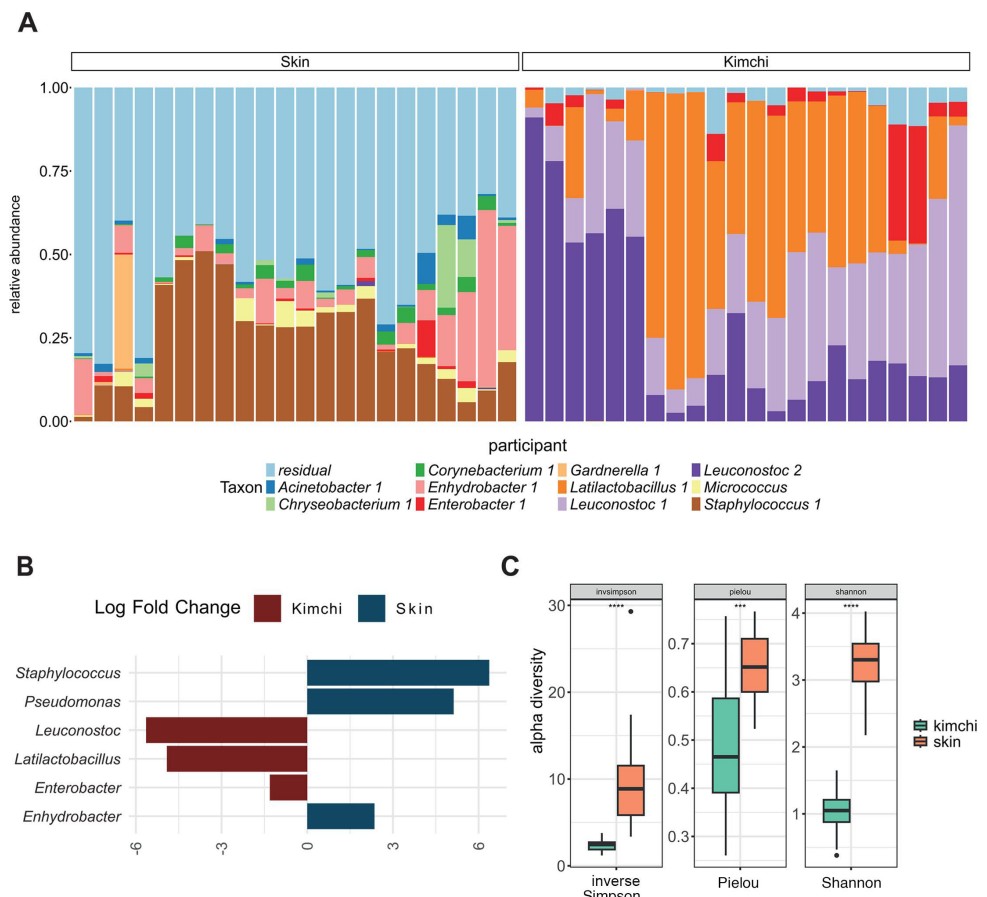

**FIG 4** (A) Comparison of the hand microbiome and kimchi microbiome after 3 days of fermentation. The 11 most abundant bacterial taxa at the ASV level are depicted in a stacked bar graph, and the remaining ASVs are grouped under residual. (B) Differential abundance of microbiota between hand skin and the kimchi microbiome. Differential abundance of genera with at least 1% mean relative abundance per environment (skin or kimchi) was calculated with ANCOM-BC2. Bars were colored based on the location where they had a significantly higher differential abundance. (C) Comparison of the alpha diversity of hand and individually made kimchi microbiome. The alpha diversity was calculated using the inverse Simpson metric, which considers the evenness and richness of the microbiota, Pielou, which is a metric for the evenness of the microbial community and Shannon which also considers evenness and richness.

Heterofermentative LAB use the phosphoketolase pathway to ferment sugars, producing not only lactic acid but also other metabolites such as ethanol and $CO_2$ (14). This could be linked to why group kimchi had a higher proportion of gas production compared with the individual kimchi fermentations. As a second major part of our analysis, we explored associations between the hand skin microbiota and kimchi microbiota for the individually prepared kimchi. Although no substantial colonization of the kimchi fermentation by the hand microbiota was observed, we did find a few taxa overlapped between the skin and the kimchi microbiome. Specifically, *Enhydrobacter, Leuconostoc, Enterobacter,* and *Staphylococcus* were present in both. Notably, *Enhydrobacter aerosaccus,* the only known species of this genus, is an aerobic Gram-negative bacterium with a salt tolerance below 1% and a minimum pH of 5 (36). These properties suggest that this species is not able to persist or grow in the kimchi fermentation environment. *Enterobacter* was detected and reached up to 20% in certain kimchi fermentations while also being present on the skin. *Enterobacter* belongs to *Enterobacteriaceae,* which are commonly found on fresh vegetables and plants and are often unwanted members of the fermentation community. We observed *Leuconostoc* in both skin and fermentation samples. *Leuconostoc* is an important member in vegetable fermentation microbial communities during early and mid-stages of the fermentations, which are commonly

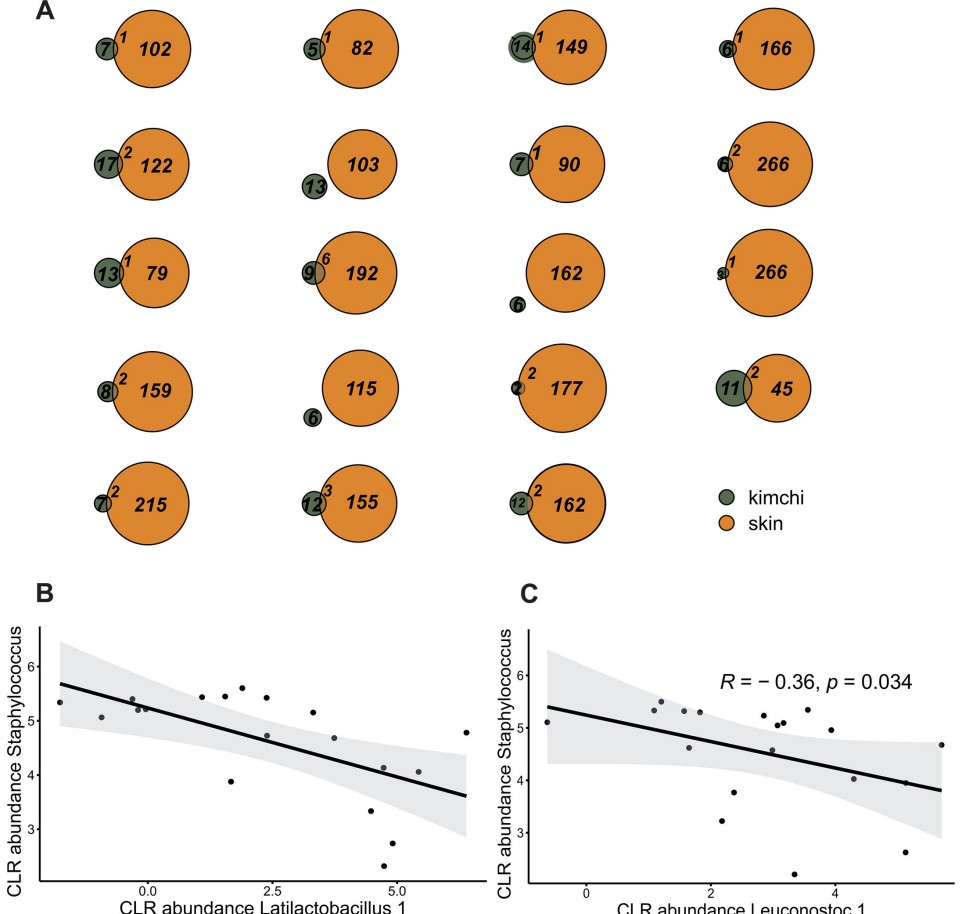

**FIG 5** (A) Overlap of bacterial taxa at the ASV level between the skin and the kimchi microbiome for individual kimchi fermentations. The occurrence of ASVs within both environments was assessed. The skin microbiome was more diverse than the kimchi fermentation ecosystem. Correlations of CLR abundance of *Staphylococcus,* present on the skin of participants, with *Latilactobacillus* 1 (B) and *Leuconostoc* 1 (C) present within the kimchi fermentations. Kendall correlation was performed on the relative abundance of *Staphylococcus* and CLR-transformed abundances of *Latilactobacillus* ASV.

characterized by the dominance of heterofermentative LAB (37, 38). *Leuconostoc* from the hands might colonize the kimchi fermentation; however, to fully substantiate this claim, in the future, strain-specific tools need to be utilized, combined with dedicated inoculation experiments to track directed colonization. This was not possible in our current citizen-science set-up and methodology of RNA-based V4-16S rRNA amplicon sequencing, in which we were only able to reliably classify up to genus level. Interestingly, we also observed an association between *Staphylococcus,* present on the hands, on LAB present in the kimchi fermentation. In particular, we found a correlation between the relative abundance of *Staphylococcus* on the hand and *Latilactobacillus* within the fermentation. Previously, a negative impact of fermentation conditions on foodborne pathogens in kimchi fermentation was shown (39). However, the effect of the presence of *Staphylococcus* from the hand on the fermentation microbiome has not been documented in the literature before. Of note, in our kimchi study, we particularly explored the directional transfer from hands to the kimchi at the start of the fermentation and not the other way around. In contrast, transfer from fermented foods to hands has been observed for fermented rice bran beds, which are stirred by hand (40). In particular, *Loigolactobacillus* was found to be present for up to 15 days on hands after contact with rice bran beds (40). In sourdough, the effect of the hand microbiome on the sourdough starter has already been shown (22). This is clearly an interesting area for future research.

The transfer from hands to fermentation is thought to be under more selective pressure. Fermentation conditions present in kimchi, such as high salinity and acidity by lactic acid, apply important abiotic selection pressures (41). We thus speculate based on our data that these fermentation conditions (high salinity and acidity) could be too harsh for the bacteria on the skin to grow and colonize the fermentation ecosystem; however, the presence of *Enterobacter* and the effect of *Staphylococcus* on the LAB present in the fermentation suggest a possible transfer of these bacteria from skin to fermentation; however, dedicated studies have to be performed to test these hypotheses. Additionally, participants were asked to disinfect their hands before the start of the workshop, as advised by the COVID-19 regulations in place at that time, which was a limitation of our study. Although this measure was mandatory and could not be altered due to the public event nature of the study, we acknowledge its potential impact. Previous studies (42, 43) have shown that although alcohol-based hand rubs do not significantly alter the overall microbial diversity of the hand skin microbiome, they can reduce microbial load and impact specific taxa proportions, which may influence the transfer of microbes during fermentation processes. Despite the indications of no meaningful direct bacterial transfer from hands to a later stage of the kimchi fermentation, further investigation during the first days of fermentation is required to understand the effect of the hand microbiome on kimchi fermentations. Our results suggest that the microbiome of the hands could potentially affect the LAB found during the first days of the kimchi fermentation. Priority effects in the gut or on plants have previously been shown to affect bacterial community composition long after the bacteria that set the stage have disappeared (44). Also, the more nuanced flavors that make different fermentations taste different may stem from the bacteria in the early stage of the fermentations.

Of note, to further explore the transfer of bacterial strains from hand to fermentation, dedicated techniques with a higher level of resolution should be utilized, such as deep shotgun metagenome sequencing, long-read sequencing, or qPCR with strain-specific primers (45). Additionally, to fully understand whether and how bacteria from the skin colonize and survive in the fermentation, dedicated inoculation experiments with skin microbiota should be performed in follow-up studies. With the current cost-efficient citizen-science set-up used in this study, only the overlap at the ASV and genus levels was reliable. Within the workshop, we aimed to control for several confounding factors: hand disinfection prior to preparation (COVID-19 regulations), identical raw material source, processing conditions (e.g., chopping), and storage conditions were all similar for both groups. Nevertheless, we were unable to control all potential confounding factors. These include differences in sample sizes between the group and individual kimchi workshops, which were determined by participant registration numbers, as well as variability in the degree of vegetable handling and processing by individual participants. Such differences could have influenced the speed and dynamics of fermentation (46).

Taken together, only a minor overlap was found between skin and kimchi microbiota, indicating that other sources are important as the origin of the fermentation microbiota. The raw ingredients are one of those important sources. LAB that are found on the fresh vegetables can grow, produce organic acids, and take over the fermentation microbial community. Additionally, the data obtained here suggest that the traditional group-based kimchi preparation influences the microbial community in a way that may contribute to a more consistent final product, particularly in terms of microbiological safety. Our citizen-science workshop underscores the importance of considering traditional fermentation practices and their potential effects on both microbial composition and the sensory characteristics of fermented foods. This community-driven approach enabled us to explore socially relevant questions and generate new hypotheses, which can be further investigated in dedicated follow-up studies. Additionally, with the help of citizens, through co-creation, we can set up more societally relevant and impactful projects in the future.

## MATERIALS AND METHODS

### Set-up of workshop and recruitment of participants

Two Kimchi workshops were designed by brainstorming between scientists and the organizers of the Sonmat festival. Two workshops were co-created and led by Ae Jin Huys, curator of the Sonmat festival (sonmat.be) and chef of Mokja (mokja.be), one kimchi workshop in which participants created an individual kimchi sample, and a second workshop in which kimchi was created in groups according to the traditional Kimjang tradition. Twenty-five participants were recruited for the individual kimchi workshop, and 20 participants for a second Kimjang group workshop. Participants provided skin microbiome samples, prepared kimchi, and also received a weck jar with kimchi that they could take home for consumption. The overall results of the study were communicated back to all participants. Experimental design and pictures are shown in Fig. 1.

### Hand microbiome sampling

At the start of the workshops, participants washed their hands in compliance with the COVID-19 restriction guidelines in place at that time. In short, participants were instructed to thoroughly wash their hands with alcogel before the activity, which also reduced the possible external contamination. The participants' hands (palm, back of the hand, and between the fingers) were swabbed using eNAT swabs (Copan, Brescia, Italy) for 30 s each before making kimchi. Prior to sampling, the swab was soaked in a vial of sterile pre-moisture buffer (50 nM Tris buffer [pH 7.6], and 0.5% Tween-20). The samples were stored in eNAT storage buffer at 4°C and transferred to −20°C for downstream processing. Each participant had one skin swab taken to match their respective individual kimchi sample. Following an explanation of the workshops, participants provided oral informed consent to have their hands swabbed. The samples were anonymized right away to prevent any participant's specific microbiota from being associated with them.

### Kimchi sampling

Kimchi fermentations were prepared with 1 kg white cabbage, 20 g salt, 80 g green onion, and 100 g onion embedded within a marinade. The marinade consisted of 50 g Korean red pepper powder, 10 g of sugar, 20 g of salt, 250 g daikon radish, 80 g pear, 50 g apple, 25 g garlic, 5 g ginger, 150 g kelp water (0.5 L water, 5 g dried kelp), and 110 g pul (consisting of 100 g kelp water and 10 g rice flour). Ingredients and vegetables were sourced from the same supplier and were identical for both workshops (individual and group kimchi). To ensure consistency across both workshops, the same chefs pre-chopped all vegetables prior to the start of the sessions. For the individually made kimchi, the participants worked independently on their own batch of kimchi. For the group kimchi, all participants worked collaboratively on a single batch of ingredients, which was then evenly divided into nine separate jars for fermentation.

During the workshop, the prepared kimchi was divided into 50 mL for microbial analysis and a weck jar that participants could take home for consumption. Individually made kimchi and group-made kimchi for microbial community analyses were stored under identical conditions in the lab, that is, for 2 days under controlled conditions at room temperature, after which the gas was released from the fermentations and moved to the fridge (4°C) for 24 h. Based on the instructions developed from the traditional practices, 48 h fermentation at room temperature was followed by 24 h in the fridge, which aligned with the instructions given to participants for home consumption. Gas production was observed visually after 3 days. On day 3, microbial cells were obtained by the addition of phosphate-buffered saline to the fermented kimchi and shaking horizontally at 1,000 rpm for 5 min. The cell suspension was further used for DNA extraction and serial dilution.

## Microbial RNA extraction and cDNA sequencing

Library prep and 16S rRNA amplicon sequencing were performed to assess the active microbial community, as previously described (28). In short, RNA was extracted using the RNeasy Microbiome Kit (Qiagen, Hilden, Germany), following the manufacturer's protocol. Leftover DNA was removed with additional treatment of DNase (Turbo DNA-free, Thermo Fisher Scientific, Waltham, MA). cDNA synthesis was performed using 806R 16S rRNA primer RNA with the Superscript III reverse transcriptase (Thermo Fisher Scientific). V4-16S rRNA PCR was performed with 10% (vol/vol) PhiX DNA (Illumina, San Diego, CA) and sequenced on the Illumina MiSeq platform using $2 \times 250$ cycles at the Center of Medical Genetics Antwerp (University of Antwerp, Antwerp, Belgium). In total, 1,033,840 reads were obtained with an average read count of 19,888 reads per sample.

## Data analysis

The obtained sequencing reads were filtered and further processed using the DADA2 pipeline (v 1.1.6) (47). In short, raw reads were filtered by removing reads with more than two expected errors and undetermined bases. Next, DADA error correction was applied using the error model, which was constructed by the alternation of sample inference and error rate estimation until convergence was achieved. Forward and reverse reads were then merged into paired contigs, and chimeras were removed. The reads were then classified from kingdom to genus or ASV level, making use of the EzBioCloud 16S rRNA database (version mtp1.5, update 2018.05) (48). Finally, ASVs classified as Archaea, Eukarya, chloroplasts, or mitochondria were removed. Further processing and analysis were performed in the R environment (version 3.5.3) in the tidyverse universe. The resulting ASV tables were processed using an in-house built package tidytacos (https://github.com/LebeerLab/tidytacos) (49). The top 12 most abundant ASVs were plotted in a stacked bar graph. Different alpha diversity metrics were calculated that take into account richness (observed, chao1), evenness (Pielou), and both (inverse Simpson and Shannon). All visualizations were made using ggplot2 (50). Significant differences were computed using Wilcoxon signed rank tests in the R environment, and correlations were performed on centered log-transformed relative abundances using Kendall correlations. Differential abundance (DA) analyses between the skin and kimchi and between the production methods of kimchi (individual versus group) were performed with the multidiff abundance R package (https://github.com/thiesgehrmann/multidiffabundance); in short, four tools were utilized to assess DA between the skin, kimchi, and kimchi methods, which are ANCOM-BC2 (51), limma (52), deseq2 (53), and lmCLR. *P*-values were adjusted for multiple testing using the false discovery rates.

## ACKNOWLEDGMENTS

The authors would like to thank all the partners of the Sonmat festival: Mokja (https://mokja.be), Vooruit Gent (https://www.viernulvier.gent/), Curieus (https://curieus.be/), and WIKIM (contribution of Kimjang tubs). The current project was funded by the European Research Council (project 852600). T.E., S.A., and S.L. were funded by the ERC project 852600. T.E. was partially funded by VLAIO (HBC.2022.1000). L.D. was funded by VLAIO Baekeland mandate (HBC.2020.2873). C.D., J.V.M., W.S., and W.V.B. were funded by grants from the Research Foundation – Flanders (FWO, 1S28622N, 1S08523N, 12ZJ821N, and 1224923N). Photographs used were taken by the photographers Willem Devriendt and Robbie Standaert.

## AUTHOR AFFILIATIONS

[1]Lab of Applied Microbiology and Biotechnology, Department of Bioscience Engineering, University of Antwerp, Antwerp, Belgium
[2]U-MaMi Excellence Research Centre, University of Antwerp, Antwerp, Belgium
[3]Curator of Sonmat Festival, Ghent, Belgium

⁴Chef and Owner of Mokja, Ghent, Belgium
⁵Faculty of Medicine and Health Sciences, University of Antwerp, Antwerp, Belgium
⁶Antwerp University Hospital, Antwerp, Belgium
⁷Centre for Environmental Sciences, Hasselt University, Hasselt, Belgium

## AUTHOR ORCIDs

Wannes Van Beeck  http://orcid.org/0000-0003-0421-8931
Tom Eilers  http://orcid.org/0000-0002-7509-2902
Wenke Smets  http://orcid.org/0000-0001-5611-6094
Sarah Lebeer  http://orcid.org/0000-0002-9400-6918

## FUNDING

| Funder | Grant(s) | Author(s) |
| --- | --- | --- |
| European Research Council | 852600 | Tom Eilers |
| | | Sarah Ahannach |
| | | Sarah Lebeer |
| Fonds Wetenschappelijk Onderzoek | | Wannes Van Beeck |
| | | Wenke Smets |
| | | Joke Van Malderen |
| | | Caroline Dricot |
| Agentschap Innoveren en Ondernemen | | Tom Eilers |
| | | Lize Delanghe |

## AUTHOR CONTRIBUTIONS

Wannes Van Beeck, Conceptualization, Data curation, Formal analysis, Investigation, Methodology, Visualization, Writing – original draft, Writing – review and editing | Tom Eilers, Conceptualization, Data curation, Formal analysis, Investigation, Visualization, Writing – review and editing | Wenke Smets, Conceptualization, Formal analysis, Investigation, Visualization, Writing – review and editing | Lize Delanghe, Investigation, Writing – review and editing | Dieter Vandenheuvel, Investigation, Writing – review and editing | Ines Tuyaerts, Investigation, Writing – review and editing | Joke Van Malderen, Investigation, Writing – review and editing | Sarah Ahannach, Investigation, Writing – review and editing | Katrien Michiels, Investigation, Writing – review and editing | Caroline Dricot, Investigation, Writing – review and editing | Nele Van de Vliet, Investigation, Writing – review and editing | Ae Jin Huys, Conceptualization, Funding acquisition, Investigation, Methodology, Project administration, Resources, Writing – review and editing | Patrick De Boever, Conceptualization, Funding acquisition, Investigation, Project administration, Writing – review and editing | Sarah Lebeer, Conceptualization, Funding acquisition, Investigation, Methodology, Project administration, Resources, Supervision, Writing – original draft, Writing – review and editing

## DATA AVAILABILITY

Sequences were uploaded to the European Nucleotide Short Read Archive under the accession number PRJEB80362.

## ADDITIONAL FILES

The following material is available online.

### Supplemental Material

**Supplemental figures (Spectrum00368-25-s0001.docx).** Fig. S1 to S6.

## Open Peer Review

**PEER REVIEW HISTORY (review-history.pdf).** An accounting of the reviewer comments and feedback.

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
