## [Reviewer comments · Microbiology Spectrum]

Microbiology Spectrum

Sonmat, a citizen-science enabled Kimjang kimchi case study on associations between hand and kimchi microbiota.

Wannes Van Beeck, Tom Eilers, Wenke Smets, Lize Delanghe, Dieter Vandenneuvel, Ines Tuybaerts, Joke Van Malderen, Sarah Ahannach, Katrien Michiels, Caroline Dricot, Nele Van de Vliet, Ae Jin Huys, Patrick De Boever, and Sarah Lebeer

Corresponding Author(s): Sarah Lebeer, Universiteit Antwerpen

Review Timeline:

Submission Date:	February 6, 2025
Editorial Decision:	March 28, 2025
Revision Received:	June 6, 2025
Editorial Decision:	August 3, 2025
Revision Received:	October 10, 2025
Accepted:	October 26, 2025

Editor: Luca Cocolin

Reviewer(s): The reviewers have opted to remain anonymous.

Transaction Report:

DOI: <https://doi.org/10.1128/spectrum.00368-25>

Re: Spectrum00368-25 (Sonmat, a citizen-science enabled Kimjang kimchi case study on associations between hand and kimchi microbiota.)

Dear Prof. Sarah Lebeer:

Thank you for the privilege of reviewing your work. Below you will find my comments, instructions from the Spectrum editorial office, and the reviewer comments.

Revision Guidelines

Sincerely,
Luca Cocolin
Editor
Microbiology Spectrum

Reviewer #1 (Comments for the Author):

This manuscript is interesting as it presents tantalizing data about the effect of human factors (group vs individual) on the composition and function of kimchi. There are anecdotal ideas that humans may be a source of small batch fermented food microbes, but there are few explorations of this. By comparing the outcomes of a group vs individual kimchi production process at a festival, this paper begins to try to do that. I appreciate how the authors are generally quite careful to not make strong

assertions about hand to kimchi transfer from their data. They instead focus on the unique aspect of using a citizen science approach to learn about kimchi microbial ecology.

With that said, I do want the authors to be even more careful with their claims. The statement "Our results suggest that the microbiome of the hands plays a role in the early successional stage of kimchi fermentation" is not supported by the study. To make this claim, you would need to experimentally separate the microbiome of the hands of people and then inoculate it into controlled microbial communities to document that 1) those hand microbes actually grow in the ferments, and 2) that they affect the fermentation. You only show some weak overlap and correlations, not causation.

One thing that is not clear to me (and is important to the overall story!) is whether the cabbage and other raw veggies used for the fermentation at the two different events (individual vs group) were the same. It has been documented that raw plant materials can impact outcomes of kimchi fermentation and fermented microbial community composition. Were the exact same vegetables used across all fermentations? How well mixed were the raw vegetables across the different ferments?

Another alternative explanation for why the group vs individual kimchi had different fermentation dynamics and communities is that they were stored differently. From what I can tell in the methods, the individual fermentations were sent home with participants, so we would expect variable temperatures for incubation. It sounds like (based on the methods, but I could be wrong) that the group ferments were incubated at room temperature in controlled conditions.

It is also unclear to me how well the processing of the vegetables in this study was controlled. The amount of chopping of vegetables can impact the release of sugars from vegetables, and that can have downstream impacts on LAB and fermentation dynamics.

All of the big caveats that are noted above should be discussed in a new paragraph in the Discussion that outlines the key caveats and limitations of this work.

Line 342: "The same approach was also suggested to participants for their sampling at home" is not clear to me. What do you mean here?

When trying to match sources to microbiomes, it is really important to control for potential contamination. Some participants had ASVs on their hands that matched kimchi ASVs. How careful were the authors with making sure these individuals did not eat a bunch of kimchi before the activity? There is not explicit discussion of blanks or negative controls in the discussion.

The presence of *Staphylococcus* in the ferments is not surprising given that it is a common contaminant of microbiome samples (from humans who process those samples). What approaches were used to make sure there was no sample contamination from human skin bacteria? Do the authors have any evidence that the very rare *Staphylococcus* in the ferments are even viable (and not just some low level DNA contaminants) from trying to plate them out to check for viability? Same is true for the *Leuconostoc* observed in both skin and ferments - is that *Leuconostoc* on the hands viable?

One minor note about the use of the term "citizen science"... some folks working in this space have noted that "community science" might be a better and more inclusive term because it does not imply citizenship in the country where the individual is helping with science. For examples, recent immigrants who are not citizens of a country could (and should!) contribute to science of that country without having obtained citizenship. I recognize that citizen science is the more commonly used term, but perhaps add something like "(or more recently rephrased as community science)" early in your paper to acknowledge this.

Reviewer #2 (Comments for the Author):

The manuscript by Van Beeck et al. is a nice example for a citizen science supported study investigating traditional fermentation practices and it interesting to see the differences in community composition between different production practices. However, there are significant gaps in the analysis and some errors in the discussion. I therefore recommend major revisions.

These were some of the major issues which need to be addressed:

- **Clarifying the Role of Citizen Science:** While this is labeled as a citizen science project, it's unclear how citizen involvement actually enabled the study. Phrases like "harnessing the power of citizen science" (L41) and "citizen-science enabled case study" seem misleading. Definitions of citizen science vary, but a key aspect is that citizens are involved beyond just sample collection-contributing to study design, data analysis, or interpretation. The listed Human Gut Microbiome Project is therefore not a citizen science project for example. I love seeing more citizen science approaches, and hosting workshops is a fantastic idea, but to put it bluntly-why couldn't a group of scientists have conducted this study themselves? I don't mean to diminish the study, but this distinction is important. I kindly ask the authors to be more mindful in their phrasing, as the current wording feels overstated.

- **Details on Fermentation Process and Sampling:** The authors should provide more justification for their sampling choices. Why

was fermentation sampled at 3 days? What was the hypothesis behind this? Additionally, why wasn't the raw material sampled? If the goal is to understand the origins of microorganisms in fermentation, this would have been crucial, as previous studies show raw ingredients as the primary microbial source. If the goal is to study the effect of these microbial communities in the unique aromas of hand made kimchi, why was this not assessed?

- Taxonomic Resolution & Study Limitations: The discussion briefly touches on taxonomic resolution, but this needs to be addressed more explicitly. Identifying taxa only at the genus level is too broad to conclude whether strains from participants' hands persisted in fermentation. This is a major limitation and should be acknowledged earlier in the paper.
- Insufficient Analysis: The analysis in this study is quite limited and doesn't fully support the claims made (e.g. of an influence of the hand microbiome on different characteristics). For publication, substantial additional analysis would be needed, e.g. showing ordination plots, testing various different diversity metrics, or performing differential abundance tests.
- Confounding Factors: The study design includes several confounding factors that aren't sufficiently discussed. These include differences in sample sizes and variations in workshop timing, both of which could significantly impact results.
- Terminology & Clarity: Some statements are convoluted or use incorrect terminology (e.g., L113, L284, L295, L303). I kindly ask the authors to revise these for accuracy and clarity.

Additional Line Comments:

- L113: The Isala project studies the vaginal microbiome-there is no such thing as a 'female microbiome.'
- L118: The word "organized" is repeated.
- L148: The sample sizes for individually made vs. group-made kimchi are very different-where are these numbers explicitly stated?
- L181: What beta diversity metric was used? Please add a table with PERMANOVA (?) results.
- L277: What does this mean? If participants were swabbed before kimchi making, this should already be clear.
- L288: This argument is unclear-do you mean bacteria from the skin?
- L291: Your results show that the same ASVs were present, but this doesn't necessarily mean they were transferred from the skin, let alone that they actively influenced fermentation conditions.
- L295: What exactly do you mean by "early successional stage"?
- L352: Please specify the primers, sequencing platform, total reads obtained, and average reads per sample.
- Supplementary File 1: Not properly labeled-what are KC1 and KC2?

We thank the reviewers for their valuable suggestions and input for improving our manuscript! Below we have included a point-by-point answer and revisions to each of the comments raised by the reviewers.

Reviewer #1 (Comments for the Author):

This manuscript is interesting as it presents tantalizing data about the effect of human factors (group vs individual) on the composition and function of kimchi. There are anecdotal ideas that humans may be a source of small batch fermented food microbes, but there are few explorations of this. By comparing the outcomes of a group vs individual kimchi production process at a festival, this paper begins to try to do that. I appreciate how the authors are generally quite careful to not make strong assertions about hand to kimchi transfer from their data. They instead focus on the unique aspect of using a citizen science approach to learn about kimchi microbial ecology.

Comment: With that said, I do want the authors to be even more careful with their claims. The statement "Our results suggest that the microbiome of the hands plays a role in the early successional stage of kimchi fermentation" is not supported by the study. To make this claim, you would need to experimentally separate the microbiome of the hands of people and then inoculate it into controlled microbial communities to document that 1) those hand microbes actually grow in the ferments, and 2) that they affect the fermentation. You only show some weak overlap and correlations, not causation.

ANSWER:

We sincerely thank the reviewer for their thoughtful and constructive feedback. We appreciate the recognition of the novelty of our citizen science approach and the careful framing of our findings. We fully agree with the reviewer's point that our current study does not establish causation regarding the role of hand microbiota in kimchi fermentation.

In response, we have revised the abstract to more cautiously reflect our findings and avoid overstatements in L42-46. As noted, the associations we observed are indeed indirect, based on overlapping microbial signatures between skin and fermentation samples, rather than direct evidence of microbial transfer or functional impact. We also acknowledge the need for follow-up studies that experimentally test the influence of hand microbiota, e.g., through controlled inoculation experiments, to determine whether these microbes can grow in the fermentation environment and influence its trajectory. We have added this

reflection to the discussion section (L282–285) to clarify the limitations of our current design and to suggest directions for future research.

Comment: One thing that is not clear to me (and is important to the overall story!) is whether the cabbage and other raw veggies used for the fermentation at the two different events (individual vs group) were the same. It has been documented that raw plant materials can impact outcomes of kimchi fermentation and fermented microbial community composition. Were the exact same vegetables used across all fermentations? How well mixed were the raw vegetables across the different ferments?

ANSWER:

We thank the reviewer for this important observation and fully agree that raw plant materials can significantly influence fermentation outcomes and microbial community composition.

To clarify: in both the individual and group kimchi sessions, the cabbages and other ingredients were sourced from the same batch and obtained from the same local provider. For the group kimchi, the vegetables were prepared and mixed collectively, ensuring a uniform starting material. In the individual kimchi preparations, participants handled and prepared their own vegetables, which may have introduced some inter-individual variation. However, this variation is inherent to the citizen science approach and reflects real-world diversity in preparation practices.

We have expanded the description of vegetable sourcing and preparation in the Methods section (L364–370) and acknowledge it as potential confounding factor in the discussion (L324–325).

Comment:

Another alternative explanation for why the group vs individual kimchi had different fermentation dynamics and communities is that they were stored differently. From what I can tell in the methods, the individual fermentations were sent home with participants, so we would expect variable temperatures for incubation. It sounds like (based on the methods, but I could be wrong) that the group ferments were incubated at room temperature in controlled conditions.

ANSWER: Thank you for raising this important point. We apologize for the lack of clarity in the Methods section regarding storage conditions.

To clarify: although participants were allowed to take their kimchi home for personal use, all samples used for microbial analysis, both from the individual and group fermentations, were collected on-site and incubated under the same

controlled conditions in our laboratory. This consistent storage approach ensures that differences in fermentation dynamics and microbial communities cannot be attributed to variation in incubation environments.

We have revised and expanded the relevant section in the methods (L374–381) to make this clearer.

Comment: It is also unclear to me how well the processing of the vegetables in this study was controlled. The amount of chopping of vegetables can impact the release of sugars from vegetables, and that can have downstream impacts on LAB and fermentation dynamics.

ANSWER: We appreciate the reviewer's comment and fully agree that the degree of vegetable processing can influence fermentation dynamics.

To clarify, in our study, the chopping and preparation of vegetables were carefully controlled and standardized across both the individual and group kimchi sessions. All vegetables were processed by the same chefs from the Vooruit kitchen, under the supervision of Ae Jin Huys, ensuring consistency in chopping size and preparation method. This controlled approach minimizes variability in sugar release and other factors that could influence microbial activity. We have updated the Methods (L364–370) and discussion section (L324–325).

Comment: All of the big caveats that are noted above should be discussed in a new paragraph in the Discussion that outlines the key caveats and limitations of this work.

ANSWER: As mentioned above, we fully acknowledge that some key information was not made sufficiently clear in the original Materials and Methods section and original manuscript. We have now revised the manuscript to clarify these points, particularly the consistency in starting material, processing, and storage across both groups. In our view, this consistency addresses the major concerns, and we would not necessarily consider these aspects to be "big caveats." Nevertheless, to reflect the importance of these issues and for full transparency, we have added a new paragraph to the Discussion section (Lines 318–331) that explicitly addresses these points and discusses their relevance as potential limitations.

Comment: Line 342: The same approach was also suggested to participants for their sampling at home" is not clear to me. What do you mean here?

ANSWER: We rewrote the section, participants received instructions on how to handle the kimchi at home for their own consumption. Microbial analysis was performed on samples (50mL) stored under controlled conditions within the lab (48h at room temperature followed by 24h at 4°C). Conditions were chosen based on instructions developed by chef Ae Jin on traditional practices. We expanded the material and methods section to make this clearer: L374-381.

Comment:

When trying to match sources to microbiomes, it is really important to control for potential contamination. Some participants had ASVs on their hands that matched kimchi ASVs. How careful were the authors with making sure these individuals did not eat a bunch of kimchi before the activity? There is not explicit discussion of blanks or negative controls in the discussion.

ANSWER:

We thank the reviewer for this important comment regarding contamination control. As noted, our workshop had open registration, so we were unable to screen participants in advance for dietary habits, including recent kimchi consumption. While we did not specifically ask about kimchi intake prior to the activity, we administered a Qualtrics questionnaire to assess participants' prior fermentation experience. Notably, 67% had never fermented vegetables before, and among the few who had, it was done only occasionally (3/5 a few times per year; 2/5 less than once per year). We have included these results as supplementary figure 5.

To minimize external contamination, participants were instructed to thoroughly wash and disinfect their hands with alcohol immediately before the activity (as was also the ongoing regulation for Covid-19 pandemic), added to methods in L353-354 and in discussion L324-325. In addition, negative controls were included during DNA extraction and PCR to monitor for potential contamination, which did not result in significant read counts. Additionally, our main finding points toward indirect effect and associations between the microbiota on the hands (*Staphylococcus* in high abundances present on the hands) and microbiota in the kimchi fermentation (*Lactobacillus* and *Leuconostoc*, both also present in high abundance in the kimchi fermentation, thus less affected to contamination). The persistence and colonization of hand microbiota in the kimchi fermentation should be examined in dedicated follow up projects with targeted inoculation experiments, which were not feasible in our current citizen science workshop set-up (added in discussion L320-324).

Comment:

The presence of Staphylococcus in the ferments is not surprising given that it is a common contaminant of microbiome samples (from humans who process those samples). What approaches were used to make sure there was no sample contamination from human skin bacteria? Do the authors have any evidence that the very rare Staphylococcus in the ferments are even viable (and not just some low-level DNA contaminants) from trying to plate them out to check for viability? Same is true for the Leuconostoc observed in both skin and ferments - is that Leuconostoc on the hands viable?

ANSWER: Thank you for these valuable remarks. To address concerns about viability and potential contamination, we employed an RNA-based approach to characterize the active microbial community. Unlike DNA, RNA degrades more rapidly under fermentation conditions and therefore serves as a better proxy for metabolically active and viable organisms. This approach has been shown to more accurately reflect the living microbial community, as previously demonstrated in our work (Wuyts et al., 2018). We adjusted the methods section with additional details for clarity (L384-394).

One minor note about the use of the term "citizen science"... some folks working in this space have noted that "community science" might be a better and more inclusive term because it does not imply citizenship in the country where the individual is helping with science. For examples, recent immigrants who are not citizens of a country could (and should!) contribute to science of that country without having obtained citizenship. I recognize that citizen science is the more commonly used term, but perhaps add something like "(or more recently rephrased as community science)" early in your paper to acknowledge this.

ANSWER: Thank you for this thoughtful comment. We are well aware of the ongoing discussion around terminology in this space — for example, we also reflected on these nuances in our recent opinion piece in Nature Medicine (<https://www.nature.com/articles/s41591-024-03371-2>). While “community science” is indeed gaining traction as a more inclusive alternative, we note that its usage is still debated. As argued by others, “community science” is best reserved for initiatives that are both driven by and directly benefit specific communities, which may not apply to all citizen science projects. To acknowledge this important distinction, we have added a brief note in the Introduction (Lines 101–106) referencing the evolving terminology.

Reviewer #2 (Comments for the Author):

The manuscript by Van Beeck et al. is a nice example for a citizen science supported study investigating traditional fermentation practices and it interesting to see the differences in community composition between different production practices. However, there are significant gaps in the analysis and some errors in the discussion. I therefore recommend major revisions.

These were some of the major issues which need to be addressed:

Comment 1: Clarifying the Role of Citizen Science: While this is labeled as a citizen science project, it's unclear how citizen involvement actually enabled the study. Phrases like "harnessing the power of citizen science" (L41) and "citizen-science enabled case study" seem misleading. Definitions of citizen science vary, but a key aspect is that citizens are involved beyond just sample collection-contributing to study design, data analysis, or interpretation. The listed Human Gut Microbiome Project is therefore not a citizen science project for example. I love seeing more citizen science approaches, and hosting workshops is a fantastic idea, but to put it bluntly-why couldn't a group of scientists have conducted this study themselves? I don't mean to diminish the study, but this distinction is important. I kindly ask the authors to be more mindful in their phrasing, as the current wording feels overstated.

ANSWER: Thank you for this thoughtful feedback. We agree that clarity around community involvement is essential. We have made it more explicit that embedding our workshop within a kimchi festival enabled participants not only to collect samples but also to contribute traditional knowledge and fermentation practices. Before the science festival with sampling, we did a cocreation session to ensure these inputs assisted the study design and interpretation, aligning with contributory models of citizen science. Thus, we continue to use "citizen science" to reflect our commitment to broad public engagement.

We agree that sometimes the term citizen sciences is coined too often. We refer to our recent publication in Nature Medicine (<https://www.nature.com/articles/s41591-024-03371-2>), where we describe a citizen science pyramid outlining different levels of engagement, from contributory to co-created research (Figure 1 added below) and referred to in the introduction (L102-114)). we also recognize the importance of deeper community participation, we have added a statement in the Discussion (Lines 337–340) acknowledging that future iterations of this work could incorporate even more participatory elements. Finally, we included additional references from the

"American Gut Project" (Line 109), to clarify the nature of participation and to more accurately reflect the scope and structure of that initiative.

These revisions aim to preserve the accurate use of "citizen science" while transparently communicating the extent of public involvement and highlighting potential for deeper engagement in future work.

Figuur 1 - <https://www.nature.com/articles/s41591-024-03371-2>

Comment: Details on Fermentation Process and Sampling: The authors should provide more justification for their sampling choices. Why was fermentation sampled at 3 days? What was the hypothesis behind this? Additionally, why wasn't the raw material sampled? If the goal is to understand the origins of microorganisms in fermentation, this would have been crucial, as previous studies show raw ingredients as the primary microbial source. If the goal is to study the effect of these microbial communities in the unique aromas of hand made kimchi, why was this not assessed?

ANSWER:

Thank you for your thoughtful comment. We have revised the Methods section (L364–384) to clarify the rationale behind our fermentation and sampling choices. The decision to sample kimchi at 3 days of fermentation was based on traditional fermentation practices, which indicate that this period typically marks the early phase when microbial communities, begin to establish and influence fermentation dynamics, especially those introduced by human handling. Our

hypothesis was that this early stage would best capture the influence of the hand microbiome before dominant lactic acid bacteria outcompete other microorganisms.

While we agree that raw ingredients are a major source of microorganisms in kimchi fermentation, our primary objective which was brainstormed by the community cocreation was to investigate the contribution of the hand microbiome to microbial dynamics and potentially aroma development. To control variation from the raw ingredients, all participants used ingredients from the same provider and batch (added in methods L364-366). We acknowledge that not sampling the raw materials limits our ability to fully trace microbial origins; however, given our specific focus on human-associated microbial contributions, we prioritized standardizing rather than characterizing the ingredient microbiota. We also acknowledge these confounding factors in our discussion (L318-L331)

We also recognize the importance of aroma as a functional outcome of fermentation and appreciate the suggestion. While aroma profiling was beyond the scope of this study, it presents a valuable direction for future work.

Comment: Taxonomic Resolution & Study Limitations: The discussion briefly touches on taxonomic resolution, but this needs to be addressed more explicitly. Identifying taxa only at the genus level is too broad to conclude whether strains from participants' hands persisted in fermentation. This is a major limitation and should be acknowledged earlier in the paper.

ANSWER:

We appreciate the reviewer's comment, and fully agree that taxonomic resolution is a key limitation of our study. This limitation stems from the use of 16S rRNA amplicon sequencing, which, while widely used and cost-effective for profiling microbial communities, typically does not allow for reliable species- or strain-level identification. We have now acknowledged this resolution more explicitly earlier in the manuscript.

To clarify, our analysis did not rely solely on genus-level comparisons. We used amplicon sequence variants (ASVs), which provide higher resolution than traditional operational taxonomic units (OTUs) and can distinguish between closely related sequences. However, taxonomic assignment of these ASVs often cannot go beyond the genus level due to limitations in current reference databases and the conserved nature of the 16S rRNA gene.

We recognize that this restricts our ability to definitively track the persistence of specific strains from participants' hands into the fermentation. Nonetheless, we

believe our approach provides a valuable foundation for generating hypotheses that can be tested in future studies using higher-resolution methods such as shotgun metagenomics or strain-resolved metatranscriptomics. we included this limitation in the discussion L318-L324

Comment: Insufficient Analysis: The analysis in this study is quite limited and doesn't fully support the claims made (e.g. of an influence of the hand microbiome on different characteristics). For publication, substantial additional analysis would be needed, e.g. showing ordination plots, testing various different diversity metrics, or performing differential abundance tests.

ANSWER: We thank the reviewer for this valuable feedback. In response, we have conducted numerous additional analyses to strengthen our findings and better support our claims regarding the influence of the hand microbiome on fermentation characteristics. These new analyses have been incorporated into both the main text and supplementary materials. Specifically, we expanded our alpha diversity analyses with various metrics focused on both richness, evenness and a combination. These analyses revealed that alpha diversity differences were primarily driven by variations in evenness. This finding has been added to the Results section (Lines 161–163), and a comprehensive comparison of different alpha diversity metrics is included in both main figures (figure 3C, 4C) the supplementary figures (figure S1, S3). We have also expanded our beta diversity analysis with ordination plots on both Bray curtis as aitchison distance (supplementary figure S4).

Furthermore, we performed an additional in-depth differential abundance analysis to compare microbial communities between individual and group fermentations, as well as between individual fermentations and skin microbiomes, using 4 different differential abundance metrics. Details on these analyses were added to the methods section “Data Analysis” L390-L413. The results of these analyses are now included in the manuscript, along with corresponding panels in figures 3 and 4 and supplementary figures. An additional differential prevalence analysis was also performed on the group versus individual kimchi samples.

We believe these additional analyses significantly enhance the robustness of our conclusions and address the reviewer’s concerns regarding analytical depth.

Comment: Confounding Factors: The study design includes several confounding factors that aren't sufficiently discussed. These include differences in sample sizes and variations in workshop timing, both of which could significantly impact results.

ANSWER: Thank you for highlighting this important point. We agree that certain confounding factors, such as differences in sample sizes and variations in the timing of workshops, could have influenced our results. We have included a discussion on the limitations in our discussion section (L317-L331) to explicitly acknowledge and reflect on these limitations.

We believe that openly addressing these limitations strengthens the transparency of our study and provides important context for interpreting the findings. These considerations also inform recommendations for future research, where more tightly controlled experimental designs could help isolate specific effects.

Comment: Terminology & Clarity: Some statements are convoluted or use incorrect terminology (e.g., L113, L284, L295, L303). I kindly ask the authors to revise these for accuracy and clarity.

ANSWER: We thank the reviewer for pointing out these inconsistencies and have revised all for accuracy.

Additional Line Comments:

- L113: The Isala project studies the vaginal microbiome-there is no such thing as a 'female microbiome.' **ANSWER: We changed the wording to vaginal**
- L118: The word "organized" is repeated. **ANSWER: we rephrased for clarity**
- L148: The sample sizes for individually made vs. group-made kimchi are very different-where are these numbers explicitly stated? **ANSWER: We have added the numbers to figure legends and method sections.**
- L181: What beta diversity metric was used? Please add a table with PERMANOVA (?) results. **ANSWER: We added the metric used (Bray Curtis dissimilarity) for PERMANOVA, and have added additional information. For PCoA betadiversity analysis for supplementary figure 2 we included both Bray-Curtis and XXX as beta diversity metric.**
- L277: What does this mean? If participants were swabbed before kimchi making, this should already be clear. **ANSWER: We revised the methods for clarity on the hand microbiome sampling (L378-381)**
- L288: This argument is unclear-do you mean bacteria from the skin? **ANSWER: We revised the sentence –**
- L291: Your results show that the same ASVs were present, but this doesn't necessarily mean they were transferred from the skin, let alone that they actively influenced fermentation conditions. **ANSWER: We agree and changed to less strong phrasing.**
- L295: What exactly do you mean by "early successional stage"? **ANSWER: changed to first days of fermentation and explicitly added that it is**

characterized by the presence of heterofermentative LAB

- L352: Please specify the primers, sequencing platform, total reads obtained, and average reads per sample. **ANSWER: Sequencing was described in previous manuscript, which we referred to, however we expanded the method section as well with requested information. (L379-388)**

- Supplementary File 1: Not properly labeled-what are KC1 and KC2? **ANSWER: revised supplementary file and included additional beta diversity metrics**

Re: Spectrum00368-25R1 (Sonmat, a citizen-science enabled Kimjang kimchi case study on associations between hand and kimchi microbiota.)

Dear Prof. Sarah Lebeer:

Thank you for the privilege of reviewing your work. Below you will find my comments, instructions from the Spectrum editorial office, and the reviewer comments. As you can read, the new information added to the manuscript needs careful discussion because they can introduce biases related to the overall aim of the study. Please address carefully what has been highlighted and introduce in the manuscript some sentences where you accept the limitations, so that the concerns are addressed. Your manuscript will have to be subjected to another round of review and if it will not be evaluated positively I will have to reject it unfortunately.

Revision Guidelines

Sincerely,
Luca Cocolin
Editor
Microbiology Spectrum

Reviewer #2 (Comments for the Author):

General:

The authors have thoroughly addressed the my previous comments and implemented most of the suggested changes. However, I have remaining major concerns - particularly regarding the newly added information that participants disinfected their hands before preparing the kimchi. In my view, this significantly weakens the study design, especially if the primary goal was to assess microbial transmission from hands to kimchi, therefore I believe that this issue will require more extensive discussion.

Citizen Science Aspect:

- I appreciate that the authors have adopted a more careful approach in phrasing and framing the citizen science aspect of their project. However, I would kindly ask them to provide a more detailed description of the initial co-creation session with citizens, as this appears to form the basis for their experimental design. Currently, the setup and outcomes of this session are not described anywhere in the manuscript.
- L112: I appreciate that the authors describe the range of possible citizen involvement, however, I would recommend avoiding explicit reference to this Tier system for citizen science, as it is not yet widely established. Instead, I suggest keeping the description contextual, specifically by clearly outlining the role that citizens played in this project (see comment above).

Grammar and Phrasing:

Some of the recent additions and edits appear to have been made somewhat hastily. I would kindly ask the authors to carefully review the grammar and phrasing throughout, as some passages are awkwardly worded or incomplete.

- L126 '[...] positioning the project between level 2 (collector) and 3 (team member) according to us.'
- L375: Additionally, participants were asked to disinfect their hands before the start of the workshop (as advised by the ongoing covid-19 regulations at that time), so hand microbiome was limited.
- L385: should be 'which are'
- Etc.

Discussion:

- L395: I question the usefulness of qPCR with strain-specific primers, given that designing even species-specific primers can be challenging. I would therefore recommend that the authors consider replacing this approach with long-read sequencing methods, which offer higher taxonomic resolution.

Sources of Microbial Diversity:

I am somewhat confused and concerned by the additional information that participants were asked to disinfect their hands before preparing the kimchi. The authors should definitively more explicitly discuss alternative sources of microbial diversity in this context. Although they acknowledged in their response to the first review that raw ingredients are a major source, this is still not addressed in the manuscript. Given the experimental setup, it is likely that microbes from the vegetables remain the primary contributors to microbial diversity in the fermentation and this should be clearly stated.

Methodology:

L173: Gas production: How this as measured? This is not described in the methods.

L352: Notably, the issue is that taxonomic classification is often only reliable up to the genus level. Therefore, stating that the analysis was "only able to classify up to ASV and genus level" is inaccurate and should be rephrased accordingly.

Figures:

- Alpha Diversity Plots Scaling: Alpha Diversity Plots Scaling: I would appreciate if Figure 3C and 4C would be split into separate graphs since the different alpha diversity metrics have different scales (e.g. richness and evenness) and only then can we better see the differences - it is not possible right now to assess this.
- Differential Abundance: These single column differential abundance heatmaps presented in Figure 3C and 4B are not really intuitive. Typical depictions include volcano plots or barcharts depicting the logfold increase/decrease. I would recommend changing these and showing e.g. ANCOM-BC2 results in the main manuscript and the other differential abundance tests in the supplementary.
- PCoA plots: Given that the skin microbiome and kimchi microbiome are likely to be distinct communities, it would be more informative to compare individually and group-produced kimchis directly. I suggest using a PCoA plot to visualize differences in community composition between these two production methods, accompanied by appropriate statistical testing to support the comparison.

Supplementary Figures:

- Check for spelling mistakes and specify the used statistical test (e.g S4. specify 'anosim')
- I do not see the Supplementary Figure 6 referenced anywhere?

On a different note, I would like to point out that the line numbers referenced in the reviewer response and those in the tracked manuscript do not match, which makes the review process more complicated and time intensive. I kindly ask the authors to carefully verify this alignment in future submissions.

Additionally, some responses to the first review appear incomplete, for example, the statement: "For PCoA beta diversity analysis for Supplementary Figure 2 we included both Bray-Curtis and XXX as beta diversity metric." I would appreciate it if the authors thoroughly review their responses to ensure clarity and completeness before resubmission.

We thank the reviewer for their valuable suggestions and input for improving our manuscript. Below, we have included a point-by-point answer and revisions to each of the comments made.

General:

The authors have thoroughly addressed my previous comments and implemented most of the suggested changes. However, I have remaining major concerns - particularly regarding the newly added information that participants disinfected their hands before preparing the kimchi. In my view, this significantly weakens the study design, especially if the primary goal was to assess microbial transmission from hands to kimchi, therefore I believe that this issue will require more extensive discussion.

Answer: We sincerely thank the reviewers for their critical and constructive feedback. The results presented in this manuscript originate from a kimchi festival held in Belgium in 2021. At that time, COVID-19 regulations were still in effect, requiring all participants to disinfect their hands using alcohol-based hand rubs.

While this measure was mandatory and could not be altered due to the public event nature of the study, we acknowledge its potential impact. Previous studies (e.g., Mukherjee et al. 2018, Kramer et al. 2025) have shown that although alcohol-based hand rubs do not significantly alter the overall microbial diversity of the hand skin microbiome, however they can reduce microbial load and affect proportions of the taxa, which may influence the transfer of microbes during fermentation processes. We have now included a reflection on this point in the discussion section to provide additional context and transparency (L262-267 in the marked manuscript with tracked changes).

Citizen Science Aspect:

- I appreciate that the authors have adopted a more careful approach in phrasing and framing the citizen science aspect of their project. However, I would kindly ask them to provide a more detailed description of the initial co-creation session with citizens, as this appears to form the basis for their experimental design. Currently, the setup and outcomes of this session are not described anywhere in the manuscript.

Answer: An initial brainstorming session was conducted with citizens, i.e. the organizers of the Sonmat Festival, during which the Kimjang Kimchi workshop took place. The recipe was developed in collaboration with Ae Jin Huys, drawing on traditional practices. The sampling strategy was designed to closely mimic household-level fermentation methods. Citizens who

participated during the workshop had a contributory role, as they provided kimchi and skin samples and were given kimchi home for consumption. The results of the study were communicated back to everyone involved. We included this description in the introduction (L113-116) and methods (L307-316, in the marked manuscript with tracked changes).

- L112: I appreciate that the authors describe the range of possible citizen involvement, however, I would recommend avoiding explicit reference to this Tier system for citizen science, as it is not yet widely established. Instead, I suggest keeping the description contextual, specifically by clearly outlining the role that citizens played in this project (see comment above).

Answer: We have rephrased the relevant sentences and removed the tier system for citizen science. Additionally, we have expanded the description of involvement in the methods section (see above).

Grammar and Phrasing:

Some of the recent additions and edits appear to have been made somewhat hastily. I would kindly ask the authors to carefully review the grammar and phrasing throughout, as some passages are awkwardly worded or incomplete.

- L126 '[...] positioning the project between level 2 (collector) and 3 (team member) according to us.'
- L375: Additionally, participants were asked to disinfect their hands before the start of the workshop (as advised by the ongoing covid-19 regulations at that time), so hand microbiome was limited.
- L385: should be 'which are'
- Etc.

Answer We thank the reviewer for noticing and their suggestions and have revised the manuscript to ensure accuracy. Additionally, we have carefully reviewed the entire text to improve grammar and phrasing throughout.

Discussion:

- L395: I question the usefulness of qPCR with strain-specific primers, given that designing even species-specific primers can be challenging. I would therefore recommend that the authors consider replacing this approach with long-read sequencing methods, which offer higher taxonomic resolution.

Answer: We acknowledge that primer design can be challenging. However, we have recently developed a pipeline that leverages pangenome data to identify unique sequences for the design of strain-specific primers. This approach enables more precise monitoring of strain engraftment across different environments. Details of this method can be found in our recent manuscript, Pangenome-based design of strain-specific primers allows the specific monitoring of engraftment in different habitats | Research Square currently under review at *NPJ Biofilms and Microbiomes* (available via

Research Square). Additionally, we also included long-read sequencing methods as alternative method for higher taxonomic resolution (L278-279)

Sources of Microbial Diversity:

I am somewhat confused and concerned by the additional information that participants were asked to disinfect their hands before preparing the kimchi. The authors should definitively more explicitly discuss alternative sources of microbial diversity in this context. Although they acknowledged in their response to the first review that raw ingredients are a major source, this is still not addressed in the manuscript. Given the experimental setup, it is likely that microbes from the vegetables remain the primary contributors to microbial diversity in the fermentation and this should be clearly stated.

Answer: We have expanded the discussion on the potential impact of alcohol-based hand rubs on the skin microbiome and have clearly listed vegetables now as a key source of microbial diversity in our concluding statement. "Taken together, only a minor overlap was found between skin and kimchi microbiota, indicating that other sources are important as origin of the fermentation microbiota. The raw ingredients are one of those important sources. LAB that are found on the fresh vegetables can grow, produce organic acids and take over the fermentation microbial community." L292-296 (in marked manuscript with tracked changes)

Methodology:

L173: Gas production: How this as measured? This is not described in the methods.

Answer: This was an observation done visually. We have included a sentence in the methods (L348).

L352: Notably, the issue is that taxonomic classification is often only reliable up to the genus level. Therefore, stating that the analysis was "only able to classify up to ASV and genus level" is inaccurate and should be rephrased accordingly.

Answer: We have rephrased the sentence.

Figures:

- Alpha Diversity Plots Scaling: Alpha Diversity Plots Scaling: I would appreciate if Figure 3C and 4C would be split into separate graphs since the different alpha diversity metrics have different scales (e.g. richness and evenness) and only then can we better see the differences - it is not possible right now to assess this.

Answer: We have separated each richness metric into its own panel with appropriately adjusted axes and have replaced the original figures accordingly.

- Differential Abundance: These single column differential abundance heatmaps presented in Figure 3C and 4B are not really intuitive. Typical depictions include volcano plots or barcharts depicting the logfold increase/decrease. I would recommend changing these and showing e.g. ANCOM-BC2 results in the main manuscript and the other differential abundance tests in the supplementary.

Answer: We thank the reviewer for their valuable recommendations. While we continue to recognize the importance of presenting results from multiple differential abundance testing methods, (as also described previously Microbiome differential abundance methods produce different results across 38 datasets | Nature Communications.), we have moved the heatmaps to the supplementary figures to streamline the main text. The results from ANCOM-BC2 have been retained in the main manuscript and are now presented as barplots in figures 3B and 4B.

- PCoA plots: Given that the skin microbiome and kimchi microbiome are likely to be distinct communities, it would be more informative to compare individually and group-produced kimchis directly. I suggest using a PCoA plot to visualize differences in community composition between these two production methods, accompanied by appropriate statistical testing to support the comparison.

Answer: We have included a new supplementary PCoA figure (Figure S2) visualizing the significant differences between individually-made kimchi and group-made kimchi. Additionally, we have also retained the comparison of all locations in figure S5.

Supplementary

Figures:

- Check for spelling mistakes and specify the used statistical test (e.g S4. specify 'anosim')
- I do not see the Supplementary Figure 6 referenced anywhere?

Answer: We have corrected the figures, specified statistical tests, and confirmed that all the supplementary figure references are included in the main text.

On a different note, I would like to point out that the line numbers referenced in the reviewer response and those in the tracked manuscript do not match, which makes the review process more complicated and time intensive. I kindly ask the authors to carefully verify this alignment in future submissions.

Additionally, some responses to the first review appear incomplete, for example, the statement: "For PCoA beta diversity analysis for Supplementary Figure 2 we included both Bray-Curtis and XXX as beta diversity metric." I would appreciate it if

the authors thoroughly review their responses to ensure clarity and completeness before resubmission.

Answer: We apologize for the inconvenience. We have verified that the line numbers are correct in the tracked version of the manuscript (with tracked changed enabled) and have ensured that all responses are complete.

Re: Spectrum00368-25R2 (Sonmat, a citizen-science enabled Kimjang kimchi case study on associations between hand and kimchi microbiota.)

Dear Prof. Sarah Lebeer:

Your manuscript has been accepted, and I am forwarding it to the ASM production staff for publication. Your paper will first be checked to make sure all elements meet the technical requirements. ASM staff will contact you if anything needs to be revised before copyediting and production can begin. Otherwise, you will be notified when your proofs are ready to be viewed.

Sincerely,
Luca Cocolin
Editor
Microbiology Spectrum